# Ultrastructure of the Olfactory Sensilla across the Antennae and Maxillary Palps of *Bactrocera dorsalis* (Diptera: Tephritidae)

**DOI:** 10.3390/insects12040289

**Published:** 2021-03-26

**Authors:** Zhao Liu, Ting Hu, Huai-Wang Guo, Xiao-Fei Liang, Yue-Qing Cheng

**Affiliations:** 1Academy of Agriculture Sciences, Southwest University, Chongqing 400715, China; 2College of Plant Protection, Southwest University, Chongqing 400715, China; TingHu991217@163.com (T.H.); g1063480666@email.swu.edu.can (H.-W.G.); liangxf22@email.swu.edu.cn (X.-F.L.); 3National Citurs Engineering and Technology Research Center, Citrus Research Institute, Southwest University, Chongqing 400715, China; 4Chongqing Academy of Agricultural Sciences, Chongqing 400723, China; cheng4195@sina.com

**Keywords:** *Bactrocera dorsalis*, olfactory sensilla, ultrastructure, antennae, maxillary palps

## Abstract

**Simple Summary:**

The environmentally friendly methods have been employed to control the serious pest, *Bactrocera dorsalis*, based on chemical communications. However, their olfaction mechanism has not been unveiled. In this study, the ultrastructure of the sensilla on the antennae and maxillary palps was examined by field emission scanning electron microscopy (FESEM) and transmission electron microscopy (TEM). The results showed that three types of olfactory sensilla (trichodea, basiconica and coeloconica) and two types of non-olfactory sensilla (chaetica and microtrichia) located on the antennae. These findings will benefit the olfactory research and the integrated management of this pest.

**Abstract:**

The sensilla on the antennae and maxillary palps are the most important olfactory organs, via which the insect can perceive the semiochemicals to adjust their host seeking and oviposition behaviors. The oriental fruit fly, *Bactrocera dorsalis* (Hendel) (Diptera: Tephritidae), is a major agricultural quarantine pest infesting more than 250 different fruits and vegetables. However, the sensilla involved in olfaction have not been well documented even though a variety of control practices based on chemical communication have already been developed. In this study, the ultrastructure of the sensilla, especially the olfactory sensilla on the antennae and maxillary palps of both males and females, were investigated with field emission scanning electron microscopy (FESEM) and transmission electron microscopy (TEM). Three types of olfactory sensillum types including trichodea, basiconica, and coeloconica, and two non-olfactory sensilla including both chaetica and microtrichia, were observed. Each of these three types of olfactory sensilla on the antennae of *B. dorsalis* were further classified into two subtypes according to the morphology and number of receptor cells. For the first time, the pores on the sensilla trichodea and basiconica cuticular wall were observed in this species, suggesting they are involved in semiochemical perception. This study provides new information on *B. dorsalis* olfaction, which can be connected to other molecular, genetic, and behavioral research to construct an integral olfactory system model for this species.

## 1. Introduction

The oriental fruit fly, *Bactrocera dorsalis* (Hendel) (Diptera: Tephritidae), is one of the most widespread and destructive agricultural insect pests in the world [1]. It has a wide range of hosts and infests more than 250 kinds of fruits and vegetables, including citrus fruits, mangos, guavas, and peaches [1,2]. The most direct economic losses caused by this pest include premature fruit drop, early rotting, and other associated crop death or damage, which, therefore, greatly reduces the production of the agricultural industry [1]. In recent years, pesticides have been used to control pest insects causing a rise in public concern. Environmentally friendly control methods are preferred to manipulate the behavior of pest insects and their natural enemies.

The antennae and the maxillary palps, covered by numerous sensilla, are primary olfactory organs. With the sensilla, the insects can perceive and accurately recognize various odorants from the surrounding environment, which trigger insect behaviors, such as host selection and oviposition decisions [3,4,5]. Furthermore, olfaction-based behavioral manipulation has been frequently used for the monitoring and prevention of oriental fruit flies. Therefore, understanding the mechanisms of the chemical communication between the environment and the oriental fruit fly has attracted considerable attention of pesticide-free pest management methods. The insect sensillum, developed from the epidermis, is a hair-like or peg-like specialized structure protruding from the antennae, and acts as a receptor for the chemical and physical stimuli in the surrounding environment. The olfactory sensilla is characterized as numerous pores on the cuticle, which permit the entry of the semiochemical contacting with the receptors on the dendrite membrane. Even though the sensillum structures have been well studied in a variety of insect species [6,7,8], previous studies have not reported the fine ultrastructure of the sensilla on *B. dorsalis* antennae and maxillary palps, despite the economic and agricultural importance of this pest.

Extensive information is available on the role of odorant-binding proteins (OBPs), chemosensory proteins (CSPs), odorant receptors (ORs), ionotropic receptors (IRs), gustatory receptors (GRs), sensory neuron membrane proteins (SNMPs), and other genes associated with the perception of host volatiles by *B. dorsalis* [9,10,11,12,13]. These studies mainly focus on olfactory receptor identification and their potential function in the perception and recognition of semiochemicals based on the odorant molecules, which can enter the sensillum via the pores on the surface of the sensillum. Compared to the extensive information on the biochemistry and physiology of *B. dorsalis*, the morphology, especially the ultrastructure of the olfactory organs, is still largely unknown. To date, few studies have been conducted on the sensillum morphology via scanning electron microscopy [14,15]. Throughout our ongoing studies on this insect, we found that existing literature on the ultra-structure of olfactory organs of *B. dorsalis* was still less well understood. For example, detailed ultrastructure of the sensilla was not considered, and the exact locations of olfactory receptor representations across the head remains controversial. Moreover, the sensillum types on the maxillary palps were understudied. Therefore, the ultrastructure of sensilla on both the antennae and maxillary palps of *B. dorsalis*, which will provide additional support for the studies of the molecular mechanisms of olfaction and chemical ecology, is vital to the continued pest management of this insect. Moreover, these sensilla can be further classified into several morphological subtypes, which may, in the future, continue to become significant if they correspond to specific electrophysiological and/or immunocytochemical studies of their function.

In this study, the external (including information on types, subtypes, abundance, and distribution) and internal morphology of the sensilla, especially the olfactory sensilla, on the antennae and maxillary palps of *B. dorsalis* were investigated and characterized with a FESEM (field emission scanning electron microscope) and a TEM (Transmission electron microscope). The results show two main types of olfactory sensilla on the antenna, including the multiporous sensilla trichodea (ST) and sensilla basiconica (SB), as well as the aporous sensilla coeloconica (SCo). These images of sensillum types display a variety of previously undescribed features, especially when compared with earlier studies of *B. dorsalis* sensilla. This study provides detailed information on the morphology of the antennae and maxillary palps, as well as on the distribution of each sensillum type across the antennae and maxillary palps. Our study will further investigations on sensorial activities that relate to the search for suitable hosts, mates, and oviposition sites within *B. dorsalis* and other members of the Tephritidae family.

## 2. Materials and Methods

### 2.1. Insects

The laboratory colony of *B. dorsalis* (originated from Fuzhou China, 2009) was reared in a cage (30 cm × 30 cm × 40 cm) at 27 ± 1 °C, 70 ± 5% relative humidity, with a photoperiod regime of 14:10 h (L:D). The adults were reared on an artificial diet containing yeast powder, honey, sugar, and Vitamin C (15:15:1, weight).

### 2.2. Field Emission Scanning Electron Microscope (FESEM)

Male and female *B. dorsalis* adults were anesthetized in a freezer (−20 °C) for 1 min. The antennae were firstly carefully removed from the head at the base and then the head was removed from the body with tiny-tipped forceps. After putting the antennae and heads (without antennae) in distilled water, they were washed three times with an ultrasonic cleaner and fixed in 2.5% glutaraldehyde for 48 h. Both separate samples were repeatedly rinsed for 3 h with 0.1 M phosphate buffer saline (PBS buffer, pH = 7.2) before being dehydrated in a graded series of 30, 40, 50, 60, 70, 80, 90, 95, and 100% ethanol solution. The samples were dried and subsequently examined with a Regulus 8100 (Hitachi, Hitachinaka, Japan) FESEM operated at 1 to 2 kV. Ten male and ten female adults of *B. dorsalis* were each examined.

### 2.3. Transmission Electron Microscope (TEM)

The sample preparation was similar to the procedure of that described above for FESEM. The heads with antennae and maxillary palps attached were fixed in the mixture solution of 4% paraformaldehyde and 2.5% glutaraldehyde for 48 h at room temperature in the dark. Afterwards, they were rinsed with 0.1 M PBS buffer, and the samples were post-fixed in 1% osmium tetroxide for 2 h at 4 °C. Next, samples were rinsed with distilled water three times. The specimens were firstly dehydrated in a graded ethanol series from 30% to 100%, followed by infiltrating with acetone 8 times for one hour and Epon 618 for 36 h at 35 °C. Subsequently, these samples were embedded in Epon 618 and polymerized at 40, 45, 50, 55, 60, 65, and 70 °C for 12 h, respectively. Ultra-thin sections of *B. dorsalis* antennae and maxillary palps were prepared using a Leica EM UC7 ultra-microtome with a diamond knife and stained with uranyl acetate (20 min, room temperature) followed by lead citrate (5 min, room temperature), and then rinsed with excess distilled water. The ultra-thin sections were examined with a transmission electron microscope (HT-7800, Hitachi, Hitachinaka, Japan) at 80 kV. Digital pictures were generated as uncompressed greyscale TIFF files, and images were obtained using a high-resolution digital camera connected to the TEM.

### 2.4. Statistical Analyses

Sensilla on the surfaces of the antenna and maxillary palps of *B. dorsalis* were identified, counted, and measured. The numbers of various sensillum types per unit surface area were measured in this study to reflect the overall sensilla density on the whole antenna (including scape, pedicel, and funiculus) as well as across the maxillary palp. Three areas on each antennal segment were randomly chosen, and the number of sensilla on the surface was counted. Counts were converted into absolute density and then analyzed with Student’s *t*-test by using the SPSS for Windows 10.0 (SPSS Inc., Chicago, IL, USA).

## 3. Results

### 3.1. General Morphology of Antenna and Maxillary Palps

The antenna of *B. dorsalis* is comprised of a scape, pedicel, and funiculus, with each constituting ~16.0, 24.5, and 59.5% of the entire antenna length, respectively. The entire antenna length of adult male *B. dorsalis* (1.40 ± 0.04 mm, mean ± S.D.) are slightly shorter than females (1.42 ± 0.03 mm). In addition, the antenna contains an arista (1.16 ± 0.32 mm) on the funiculus (Figure 1A). Several pores were observed on the pedicel segment surface, near the sensilla chaetica (mentioned below, Figure 1C). Overall, the morphology of the male and female antenna is quite similar.

The maxillary palps are located at the base of the rostrum of *B. dorsalis*. At rest and during flight, the maxillary palps hide between the retracted proboscis and the head capsule. The maxillary palp length has no significant difference between males and females (Figure 1B).

Olfactory sensilla (OS) and nonolfactory spinules (NS) compose the majority of dense hairs covering the antennae (Figure 1D,E) and maxillary palps of *B. dorsalis*. In total, there are five types of sensilla on antennae and maxillary palps, including the olfactory sensilla: sensilla basiconica, sensilla trichodea, sensilla coeloconica, and non-olfactory sensilla: sensilla chaetica, microtrichia on the antennae. On the maxillary palps, sensilla basiconica, and non-olfactory microtrichia sensilla chaetica were observed. These sensilla types are classified according to their morphological shape, length, and width. The relative abundance and distribution of each of these identified sensillum types are different across olfactory structures (Table 1). Sensilla chaetica, sensilla trichodea, sensilla basiconica, and microtrichia are classified as single-walled sensilla, and while sensilla coeloconica are classified as double-walled sensilla (Steinbrecht, 1996).

### 3.2. Sensilla on the Antenna and Maxillary Palp

#### 3.2.1. Sensilla Trichodea

The sensilla trichodea (ST) are the most abundant type of olfactory sensilla on the antennae of *B. dorsalis*, which are distributed across the whole antennal funiculus (Figure 2A). ST are elongated, straight, or slightly curved. The hair shaft proper gradually tapers into a sharply pointed tip (Figure 2B). The length of ST on the male antennae is about the same as the sensilla on the female antennae. There are numerous pores on the surface of ST (Figure 2C), and thus, we classified them into olfactory sensilla. There are no ST on the surface of maxillary palps.

The antennal ST exhibited a single, thick, cuticle wall that is penetrated by multiple nanoscale tubules, which are distributed irregularly and also verified in TEM analyses (Figure 2D,E). Two subtypes of these sensilla were classified as ST-I and ST-II, which are innervated by one and two olfactory receptor neurons, respectively. The dendrites of the receptor neurons extend towards the sensillum lumen without branches, but it may be an exception to the dendrites near the tip of the sensilla. Individual dendrites contain different numbers of microtubules. The tubular pores are adjacent to the sensillum lumen through electron-dense material, pore tubule, which gradually widen towards the endocuticle layers.

#### 3.2.2. Sensilla Basiconica

The sensilla basiconica (SB) are round tip sensilla with a flexible socket or inflexible socket. There, numerous wall pores are observed on the cuticle, with pore tubules of 19–25 nm diameter extending into the sensillum lumen. The thickness of SB wall is nearly equivalent throughout the length of the sensillum shaft. No significant differences were detected in length of SB between males and females.

Based on the sensillum size and shape, the SB on the antennal funiculus are classified into two subtypes: antennal sensilla basiconica I (AnSB-I, Figure 3A) and antennal sensilla basiconica II (AnSB-II, Figure 4A). The AnSB-I are tapered from the base towards the blunt tip, while AnSB-II are much thinner and flatter. The SB-II was significantly longer than AnSB-I; the average length of AnSB-I is 5.6–9.8 μm, while the average length of AnSB-II is 6.2–13.1 μm. The density of wall-pores on AnSB-I (21–24 pores/μm^2^) is lower than that on AnSB-II (38–45 pores/μm^2^). The diameters of the circular wall-pores of AnSB-I and AnSB-II are different. They measure 42.6 to 52.1 nm (*n* = 20) of AnSB-I and 31.3 to 33.3 nm (*n* = 21) of AnSB-II. The wall-pores appear to be arranged longitudinally along the sensillum, and the average distance between the pores is about 68 nm on AnSB-I and 43 nm on AnSB-II (*n* = 10 pairs of pores measured from each sensillum). However, the distribution of the pores at the sensillum base is less organized than that on the distal portions of the sensillum.

From the TEM images, the cuticular wall of both sensilla basiconica is 90–110 nm thick and contains pores of 37–41 nm width. The pores of SB are connected to the sensillum lumen through tubules. In cross-section analyses of sensillum structure, either two or four receptor cells innervate AnSB, producing various types of numbers of dendrites (Figure 3D,E), which may divide into many dendritic branches (Figure 3F) within each sensillum. The neuron dendritic branch is restricted to a small region of the sensillum, namely the structure that is like a brush (Figure 3B). The outer dendritic segment appears as one dendrite at the basal part of the sensillum (Figure 3C). Some of the SB exhibited lamellate dendritic branches (Figure 4B), whereas others showed varying sizes of circular dendritic branches (Figure 3F).

Only a singular type of sensilla basiconica was found on the maxillary palps of *B. dorsalis* (MaSB). This type of sensillum is similar to AnSB-I in morphology (Figure 4C). The difference between these two kinds of sensilla is that AnSB-I is a straight or slightly curved at the tip, while MaSB curves at a quarter of the sensillum from base, and thus, are more curved overall. The length of MaSB was 6.3 to 7.7 μm, which is similar to AnSB-I. However, the width of the MaSB base is 1.5 to 1.8 μm, and is, therefore, consistently wider than AnSB-I. The density of the wall-pores on MaSB is 29.2 to 32.4 pores/μm^2^, and the pore diameters are 36.2 to 41.0 nm (*n* = 20). Thus, the density of pores on MaSB is comparable to AnSB-II while pore diameter is comparable to those pores observed on AnSB-I.

#### 3.2.3. Sensilla Coeloconica

The sensilla coeloconica (SCo) are mainly distributed in the middle of the funiculus (Figure 5A) and the density of SCo is relatively lower than other kinds of sensillum (Table 1). The SCo arise from a socket and there is plurality of protruding ribs gathered from the base toward the tip into a “finger” shape on the sensillum surface (Figure 5B). These sensilla are the shortest sensillum on the surface of the antennae. The length of sensilla coeloconica is 1.35 to 2.8 μm, with a width of 0.75 to 1.27 μm at the base. No wall-pores were found on the cuticle surface. This kind of sensilla was supposed to be another kind of olfactory sensilla because the finger-like projections own the potential spaces, aiding the odorant to penetrate the cuticle. We found that SCo are only distributed on the antennae, and no distribution was found on the maxillary palps.

With the analysis of the cross-section, it is obvious to find that the SCo is composed of the double-wall (Figure 5C,D). The thickness of inner and outer cuticle ranges from 17 nm to 24 nm of the inner one and 78 to 91 nm of the outer one, respectively. Electron dense tubules and wax canal filament-like structures filled the space between these two cuticular walls. Petal-like structures were observed in the cross-sections throughout the distal part of the sensilla. The outer walls are covered by the grooves, and the inner walls appear to be fused together. Two or three receptor neurons innervate this type of sensilla in *B. dorsalis*; thus, we classified this kind of sensillum into two subtypes, SCo-2 and SCo-3. Usually, there are two or four dendrites in the sensillum lumen. Interestingly, one of neuronal dendrites may terminate before reaching the tip of SCo-2 and SCo-3.

#### 3.2.4. Non-Olfactory Sensilla

In addition to these olfactory sensilla, the non-olfactory sensilla, including sensilla chaetica (SCh) and microtrichia (MI), were also observed on the antennae and maxillary palps of the oriental fruit fly.

SCh are the longest sensillum observed on the antennae, and are mainly distributed on the scape and pedicel (Figure 6A). SCh on male antennae are slightly longer than that on the female antennae, and have a wider range of base diameter variations (1.6 to 8.3 μm). This kind of sensilla is variable in length (27 to 150 μm) but consistent in morphology, including a pointed tip end, longitudinal grooves on the cuticular wall (Figure 6C), and basal insertions that produce a socket (Figure 6D). Despite large differences in size, all SCh are of the same thick-walled aporous type (Figure 6E). The bristles have a thick cuticular wall, and their lumen is devoid of neuronal dendritic. Thus, it appears that there are neither pores on the sensillum surface nor that the olfactory nervous system innervates the sensillum. Sensillum morphology on the maxillary palps is similar to SCh on the antennae, and the length of this sensillum is relatively steady (length = 33 to 46 μm; base diameter = 2.8–3.9 μm) (Figure 6B).

The microtrichia (MI) are the most abundant sensilla on the antennae and maxillary palps. These MI are slightly curved along their length with sharp tips and longitudinal grooves along the shaft (Figure 7A,B). This kind of sensilla is less dense on the male flies than that on the female pedicel, but it is more abundant on the male maxillary palps when compared to those on the female maxillary palps. The longitudinal- and cross-sections show the nonporous cuticular wall and the absence of dendrites, with the sensillum lumen inside (Figure 7C,D). Therefore, these sensilla are non-olfactory based on the morphology structure, the absence of the cuticular pores, and the neuronal dendrites.

## 4. Discussion

Insect behaviors are mainly influenced by chemical cues in the air. The antennae and maxillary palps are the most important olfactory organs for most insects, which numerous olfactory sensilla distribute on. The ultrastructure of various sensilla on the antennae and maxillary palps of *B. dorsalis* were investigated in this study. It was found that the antennae and maxillary palps were covered with numerous olfactory sensilla (i.e., ST, SB, and SCo) and non-olfactory sensilla (i.e., SCh and MI). The types, abundance, and distribution of these sensilla are similar to that observed from other Dipteran species, such as *Anastrepha serpentine* [16], *Haematopota pandazisi* [8], and *B. zonata* [17]. These findings were confirmed by previous research [18], except the ST, which we classified into olfactory sensilla based on the multi pores on the surface of the sensilla, observed with the FESEM. No differences were found about the morphology, abundance, and distribution of the same sensillum subtype between male and female on the antennae or on the maxillary palps. However, we could not conclude that the same morphological types of sensilla exhibit the same olfactory functions between males and females in *B. dorsalis* without more evidence.

ST is the most abundant sensilla on the antennal funiculus of the male and female oriental fruit fly. These sensilla are single-walled and multiporous. The cuticular wall is thicker than others, which helps the sensillum maintain the pore structure in the air [19]. These findings morphologically coincided with the sensilla of other Dipteran families [16,20,21,22], which are well documented as olfactory sensilla. Recently, it was reported that this kind sensilla own the function in plant volatile and pheromone detection [23]. Thus, we conclude this type of sensilla in *B. dorsalis* are olfactory sensilla based on the above information.

The SB are mainly distributed on the antennae and maxillary palps, which have been reported in various species such as *Hydrotaea chalcogasten* [21], *Toxotrypana Curvicauda* [22], *Pseudacteon tricuspis* [24], and *H. pandazisi* [8]. They are identified as olfactory sensilla with ultrastructure characteristics of numerous nano pores present on the surface. Odorant binding proteins, chemosensory proteins, and odorant receptors, which are involved in the detection of semiochemicals, have been reported in *Microplitis mediator* [25], *Adelphocoris lineolatus* [26], and *Schistocerca greraria* [27]. Some OBPs have been reported binding with plant volatile or pheromones. When the Obp mRNA in SB was deleted, the *Drosophila* electrophysiological response to selected volatile and the oviposition behavior was altered [28]. In *B. dorsalis*, there are numerous nano pores on the surface, suggesting that they are involved in the olfactory function as in *Drosophila* and *Phoracantha semipunctata* [29,30]. It is reasonable to deduce that different subtypes SB are responsible for different odorant detection, even though more evidence from electrophysiology and behavior evidence are needed.

Sensilla coeloconica (SCo) can be easily distinguished from the other sensilla due to their unique morphology, and the grooves between the cuticular fingers. This structure was confirmed in *B. dorsalis*, and was consistent with the SCo structure observed on the olfaction organs of *D. suzukii*, *Megaselia scalaris,* and *Manduca sexta* [27,31,32]. However, the grooves are not detectable in some previous research [15]. We attribute this to the electron microscopy that we used in this research, which was field scanning electron microscopy, and other researchers [27,31,32]. The SCo is involved in olfaction, and shown electrophysiology in several species. In *Drosophila*, the olfactory genes, including *Or*, *Gr*, and *Ir* genes, are expressed in SCO [33], and these genes are essential in ammonia, amines, and acids perception examined by single sensillum recording experiments [34]. Thus, we classified this kind of sensilla as olfactory sensilla. However, in *Aedes aegypti*, Sco was thought to be thermo- and hydro-sensillum because of the humidity and temperature receptor cells in this sensillum [35]. Based on the above information, the SCO in *B. dorsalis* probably function in the semiochemical sensation, but more electrophysiological and behavioral evidence are necessary.

The other types of sensilla identified both on the antennae and maxillary palps of *B. dorsalis* were sensilla chaetica and sensilla microtrichia. They both display all the characteristics of non-olfactory sensilla when compared to other well-studied Dipteran models. The SCh thick walls and no pores are similar to those of other Tephritidae fruit flies such as *B. zonata* [17]. This kind of sensilla contact the surrounding objects nearby with the antenna based on their distribution and length. Therefore, they are hypothesized to function in sensing mechanical stimuli, or in providing protection for the antennae. The sensilla chaetica on the maxillary palps also have a similar mechanical function, which has been reported in various insects including the *Drosophila*, *Neobellieria bullate,* and *Hydrotaea chalcogaster* [21,36,37]. The distribution of microtrichia (MI) on *B. dorsalis* maxillary palps is similar to that of Dipteran families, such as *M. domestica*, N. bullate, and *H. chalcogaster*, and were hypothesized to own mechanical function [21,37,38].

## 5. Conclusions

The *B. dorsalis* are able to perceive the semiochemicals in the surrounding environment via olfactory sensilla to regulate the various behaviors including host selection and oviposition localization. The current research describes the ultrastructural morphology of the sensilla on *B. dorsalis* antennae and maxillary palps in detail, using both FESEM and TEM. The olfactory sensilla are mainly distributed on the antennae and the maxillary palps including the sensilla trichodea, sensilla basiconica, and sensilla coeloconica because of the multipores on the cuticle of these sensilla. The non-olfactory sensillum chaetica and sensillum microtrichia are also identified based on the lack of pores on the cuticle of the sensilla. This information will benefit the *B. dorsalis* olfaction research by providing the sensilla morphology on ultrastructure.

## Figures and Tables

**Figure 1 insects-12-00289-f001:**
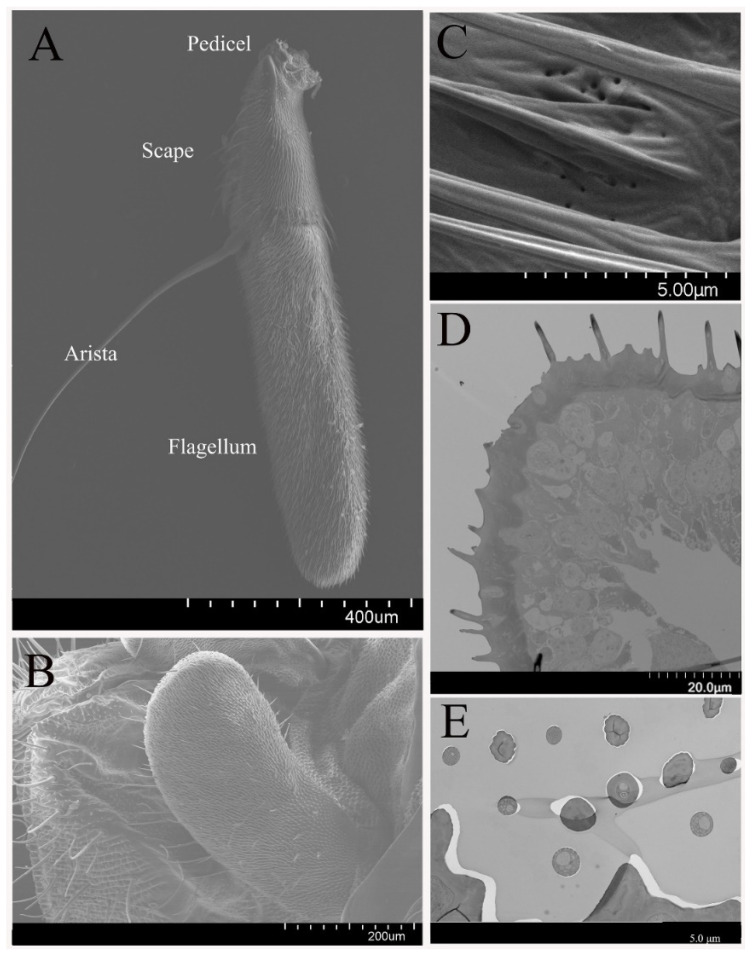
Morphology of *B. dorsalis* antenna and maxillary palps. (**A**) Numerous sensilla distributed on female *B. dorsalis* antenna, including the scape, pedicel, funiculus, and arista. The distribution on male antenna is similar to the female. (**B**) Numerous sensilla on male *B. dorsalis* maxillary palp. The sensilla distribution on female maxillary palp is similar to the male. (**C**) Pores on the pedicel surface near the sensilla chaetica. (**D**) Cross-sections at the base and the tip (**E**) of the antennae show the distribution of various sensillum types.

**Figure 2 insects-12-00289-f002:**
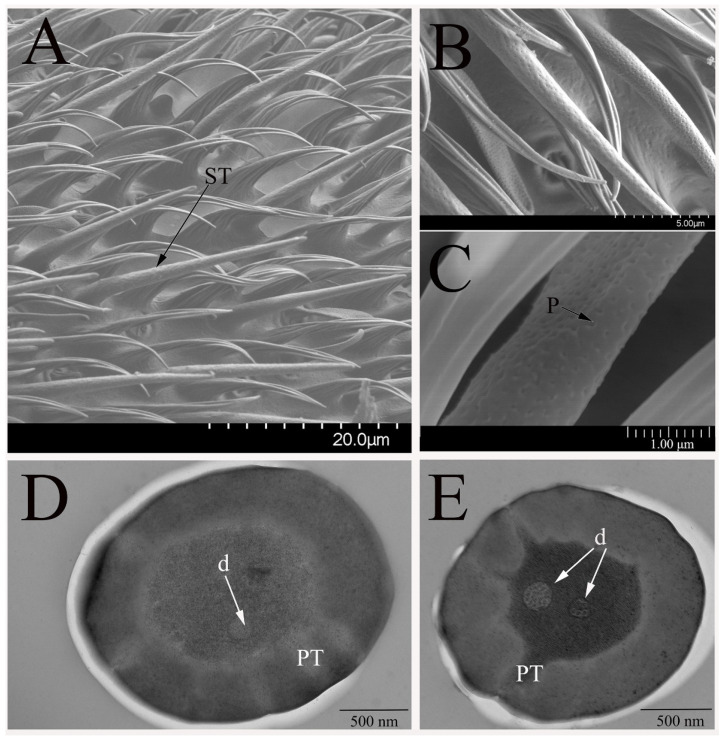
Sensilla trichodea (ST) on *B. dorsalis* antenna. (**A**) The regular array of ST on male adult antennal funiculus. (**B**) Higher magnification of shaft surface of ST on male antennae. (**C**) Showing the ultrastructure of the high density distributed tubular pores (P) on the cuticle surface. The ultrastructure of ST on female antennae is similar to that of male antennae. (**D**) ST subtype I (ST-I) contains one dendrite (d). (**E**) ST subtype II (ST-II) contains two dendrites. The numerous pores on the cuticle surface and the pore tubules are surrounded by the sensillum lymph.

**Figure 3 insects-12-00289-f003:**
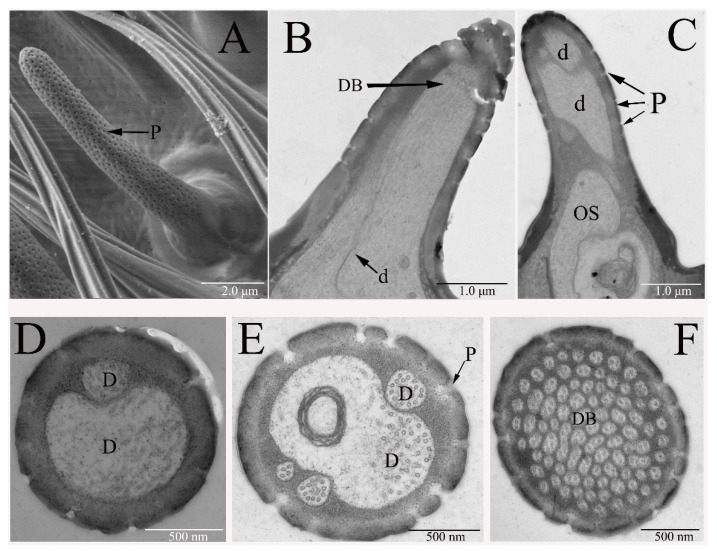
Sensilla basiconica subtype I on *B. dorsalis* antennae (AnSB-I): (**A**) AnSB I with conspicuous pores on male antenna. (**B**) A single dendrite (d) is divided into many dendritic branches (DB). (**C**) Longitudinal cross-section of AnSB-I on male adult antenna shows the outer dendritic segment (OS) at the basal part of the sensillum. (**D**,**E**) Cross-sections of AnSB-I mid-region show different numbers of dendrites in the lumen of a male antenna. (**F**) Cross-section of AnSB-I tip shows many dendritic branches divided by dendrites. The tubules with pores are observed on the cuticle. The ultrastructure of AnSB-I on the female antenna is same as that of the males.

**Figure 4 insects-12-00289-f004:**
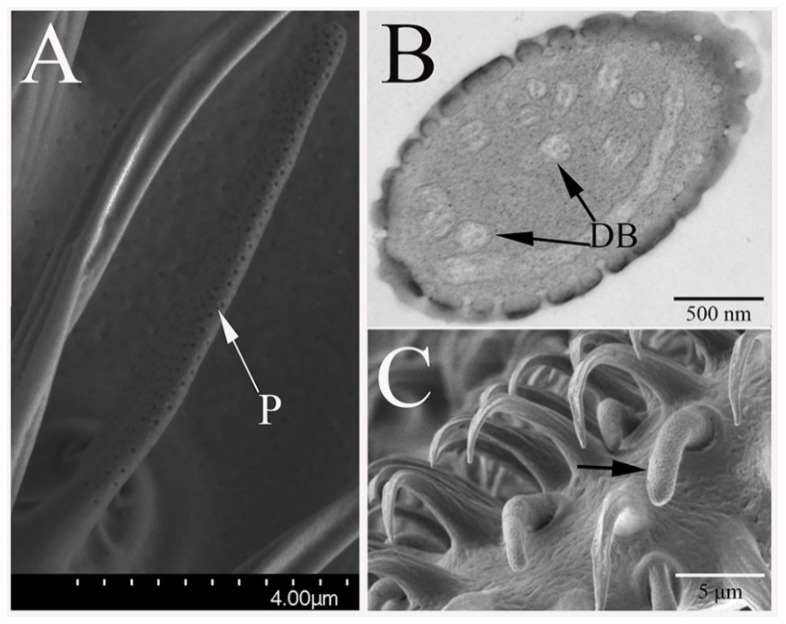
Sensilla basiconica subtype II on *B. dorsalis* antennae (AnSB-II): (**A**) the morphology of AnSB-II with conspicuous pores on male antennae. It is much thinner than AnSB-I. (**B**) Various dendritic branches are embedded in the AnSB-II lumen. The dendrite has a lamellar and a circular structure. (**C**) Sensilla basiconica on male maxillary palp (MaSB, black arrow marked). No morphology difference is found between the male and the female.

**Figure 5 insects-12-00289-f005:**
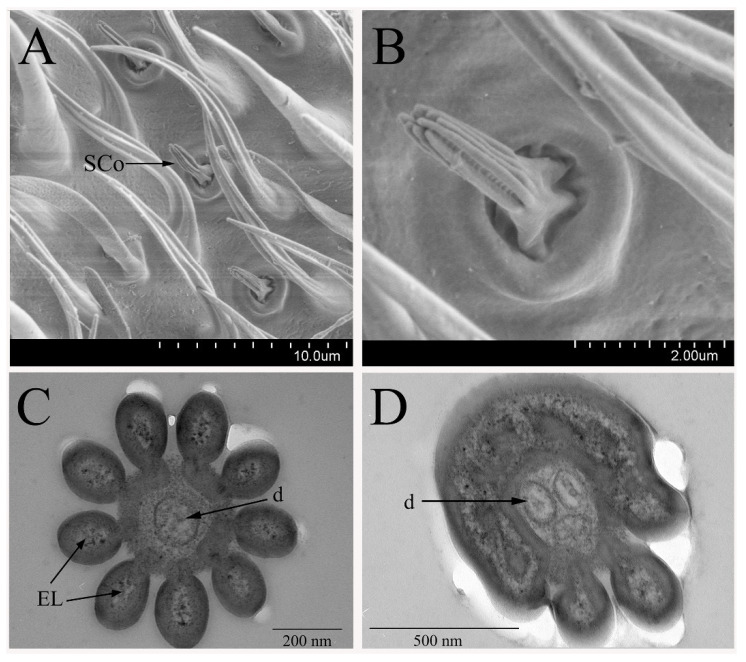
Sensilla coeloconica on the flagellum of *B. dorsalis*: (**A**) distribution of SCo on the funiculus. (**B**) SCo on a male adult antenna, showing the “finger-like” structures. (**C**) Cross-section of the distal region of the SCo shows the central lumen is surrounded by several petal-like extra-lumen (EL). One dendrite (d) is visibly accessible because the other neuron’s termination is covered by the visible plane. (**D**) Cross-section of SCo-3 on male antennae shows three dendrites appear in the sensillum lumen. The double-wall of the sensilla is visible, and electron dense tubules fill the space between the outer and inner walls.

**Figure 6 insects-12-00289-f006:**
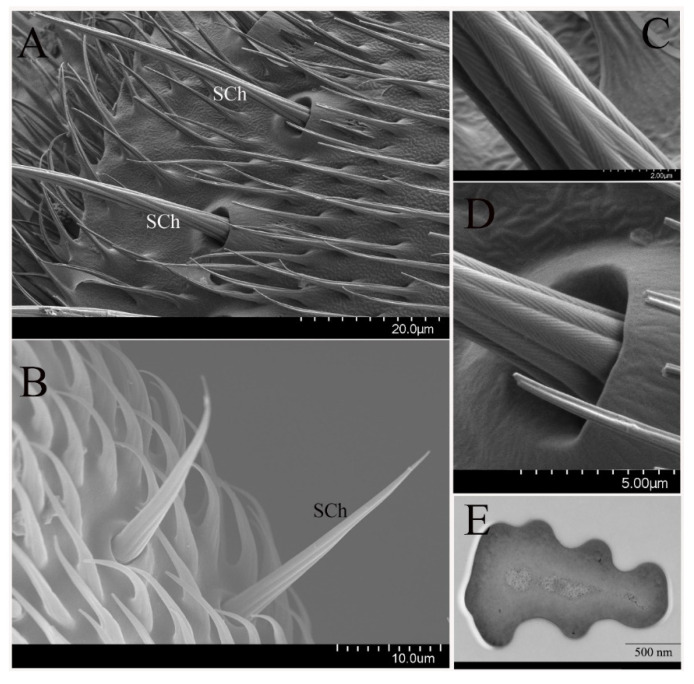
Sensilla chaetica (SCh) on *B. dorsalis* antenna (**A**), and maxillary palps (**B**), with pointed tips. (**C**) High magnification of longitudinal grooves lacking cuticular pores. (**D**) The morphological details of SCh surface showing the distinct basal socket. (**E**) Cross-section of SCh along the antenna show the thick cuticular wall, and no neuronal dendritic in the lumen.

**Figure 7 insects-12-00289-f007:**
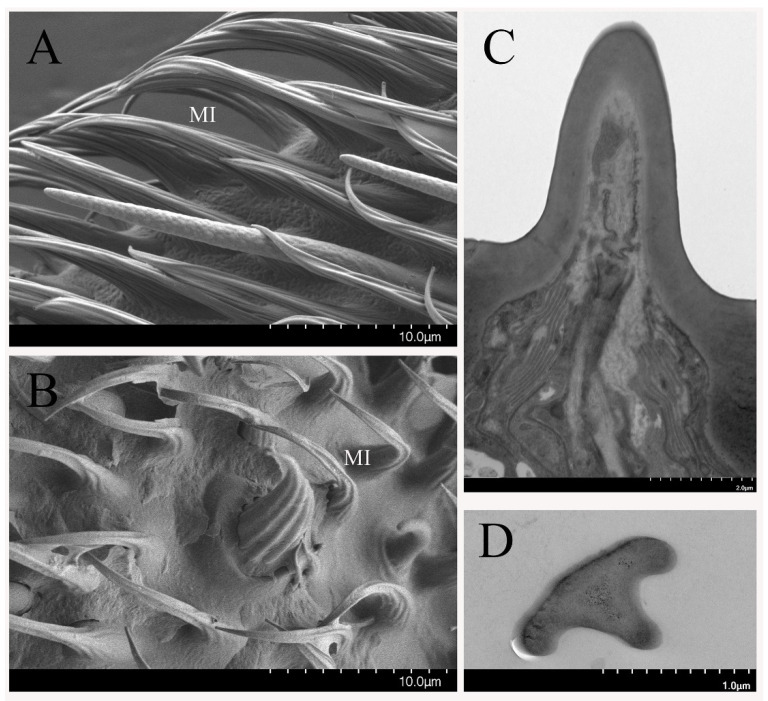
Microtrichia (MI) on *B. dorsalis* antennae (**A**,**C**,**D**) and maxillary palps (**B**). (**A**,**B**) Overview image shows the sharp tips and longitudinal grooves on the MI surface. (**C**) Longitudinal section of MI without cuticular pores and dendrites. (**D**) Cross-sections from the distal of this kind of sensillum.

**Table 1 insects-12-00289-t001:** Abundance and distribution of sensilla on *B. dorsalis* antennae and maxillary palps.

Sex	Antenna							Maxillary Palps		
	Scape		Pedicel		Flagellum					
	St	Sc	St	Sc	St	Sb	Sco	Sc	St	Sb
♂	321 ± 13	19 ± 1	793 ± 30	70 ± 5	251 ± 17	90 ± 5	12 ± 1	10 ± 2	501 ± 15 *	70 ± 9
♀	305 ± 17	20 ± 1	866 ± 36 *	68 ± 5	250 ± 10	87 ± 6	12 ± 1	10 ± 2	421 ± 28	69 ± 9

Values are mean ± S.D. for the number of different types of sensilla on antenna and maxillary palpus of *B. dorsalis*; the values of St, Sc, Sb, and Sco are mean density number per unit area of 10,000 μm^2^ on the surface of the antenna and maxillary palpus. St, Sensilla trichoidea; Sc, Sensilla chaetica; Sb, Sensilla basiconica; Sco, Sensilla coleoconic. Significant differences of the sensilla on male and female were determined using Student’s *t*-test (* *p* < 0.05).

## Data Availability

Not applicable.

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
