# Peer review of "Ultrastructure of the Olfactory Sensilla across the Antennae and Maxillary Palps of Bactrocera dorsalis (Diptera: Tephritidae)"

_insects, 2021, doi:10.3390/insects12040289_

Round 1

Reviewer 1 Report

The manuscript " Ultrastructure of the olfactory sensilla across the antennae and  maxillary palps of Bactrocera dorsalis (Diptera: Tephritidae)" by Liu et al is a nice work. This study examined B. dorsalis adult antennae, maxillary palps and sensilla on these two tissues by using FESEM and TEM. The methods are appropriate and the figures are well-prepared. Three types of olfactory sensilla and two types of non-olfactory sensilla were observed and described. It increases the knowledge on B. dorsalis olfactory system and may help the future investigations. The project were conducted soundly. The only concern is about English writing. There are quite a few spelling or grammar errors, which have been labeled in the file I attached. I suggest a further proof-reading or revision from a native-speaker will help this manuscript be published.

Author Response

Thanks. We have revised the manuscript as your suggestions.  Please see the attachment.

Reviewer 2 Report

Very nice piece of work.

My issues are with the introduction and general writing: Sentences are not always clearly written with some verbs or prepositions missing. Some sentences are also very repetitives. Read through again to correct these. Changes in the introduction are most needed: focus on defining the structure of the peripheral olfactory system, the sensilla, the neurons and their roles in semiochemical perception of various nature pheromone, host odours, acid amine etc, (as you did discuss it). It is not necessary to put too much emphasis on the role in pest management throrough the introduction: within the first sentences is sufficient, then focus on the actual scientific value of having these structure described (eg. for filling knowledge gaps which will be very valuable to study single neuron perception of chemicals in this fly)

Below my comments while reading

l18: the antenna and maxillary palp are the organs, the sensilla are prolongated cells hosting the sensory neurons. 

l43-45: these sentences are coming too early as the terms "semiochemicals" "chemical communication" are defined after. Perhaps rephrase or move lower in the introduction in l51 for example where it would fit better.

l46: integrative management systems is also unclear to most. Could be define or replace with sustainable or pesticide-free pest management methods.

l51 sensilla is too undefined. especially for this paper about its structure it is very important to well define what a sensilla is, as described in other species. 

l68-70 instead of pejorative terms such as insufficient and understudied, I would suggest to rephrase into something more into: not yet studied? or still to be determined? 

l130. Why take density instead of counts? did you check the data distribution? is the test chosen fitting the data? 

figure 1: very nice pictures! 

table 1: significant differences between what and what? 

l185 here is a mention of innervating neurons. This should have been introduced first. you need to explain clearly in the introduction what a sensilla is and what are the structures of the sensilla you will look for and described: external aspect, internal structure (hence the composition in neurons which is the essential part of sensilla). you can use the model fly Drosophila or a general statement of the structures found accross insect species. Then narrow down to the specific cheracterisation you are doing in B dorsalis. To finish, discuss how it varies with other species (as you did).

figure 2: very nice pictures and overall good and clear description of the structures observed.

l201: olfactory sensilla? I am not sure what you mean because the trichoids are also olfactory.

figure 3&4: very nice and again well described. However avoid saying it is "obvious"as it is not for people that are discovering! 

figure 5: very nice pictures. I am curious about what is this "pore-like hole' in which the sensilla is positioned? there are so different from the others!

figure 6: very nice and very curious? there are no neurons in these Sch and MI Should they be called sensilla? or have a complete different name as they are a different structure? 

l355 need rephrasing: what is the difference between this study and previous? 

l358 A rephrasing is needed. Olfactory receptor genes or, gr and ir or olfactory receptor ORs GRs and IRs.  In the introduction these are again not well defined. if not needed in the introduction,  they should be completely defined here (olfactory, gustatory, ionotropic receptors) with their role and presence in the distinct sensilla, as described in other species (very well in Drosophila).

l362 that is interesting! there could be several function to this single sensilla type???

Author Response

Dear reviewer:

Thanks for your time and valuable suggestions for this manuscript. We have carefully read your comments and made proper revsions, and plase see the atachement.  We hope the writting is acceptable for you now. We greatly appreciate your effort for improving this manuscript.

Sincerely,

Zhao Liu and the authours.
